# Psychological Resilience to Suicidal Experiences in People with Non-Affective Psychosis: A Position Paper

**DOI:** 10.3390/ijerph19073813

**Published:** 2022-03-23

**Authors:** Patricia A. Gooding, Kamelia Harris, Gillian Haddock

**Affiliations:** 1Division of Psychology and Mental Health, School of Health Sciences, Manchester Academic Health Sciences Centre, University of Manchester, Manchester M13 9PL, UK; patricia.gooding@manchester.ac.uk (P.A.G.); gillian.haddock@manchester.ac.uk (G.H.); 2Greater Manchester Mental Health NHS Foundation Trust, Manchester M25 3BL, UK

**Keywords:** resilience, suicide, suicidal experiences, psychosis, psychological interventions

## Abstract

It is important to understand the psychological factors which underpin pathways to suicidal experiences. It is equally as important to understand how people develop and maintain resilience to such psychological factors implicated in suicidal experiences. Exploring optimal routes to gaining this understanding of resilience to suicidal thoughts and acts in people with severe mental health problems, specifically non-affective psychosis, was the overarching aim of this position paper. There are five central suggestions: 1. investigating resilience to suicidal experiences has been somewhat over-looked, especially in those with severe mental health problems such as schizophrenia; 2. it appears maximally enlightening to use convergent qualitative, quantitative and mixed research methods to develop a comprehensive understanding of resilience to suicide; 3. relatedly, involving experts-by-experience (consumers) in suicide research in general is vital, and this includes research endeavours with a focus on resilience to suicide; 4. evidence-based models of resilience which hold the most promise appear to be buffering, recovery and maintenance approaches; and 5. there is vast potential for contemporary psychological therapies to develop and scaffold work with clients centred on building and maintaining resilience to suicidal thoughts and acts based on different methodological and analytical approaches which involve both talking and non-talking approaches.

## 1. Background and Overview

This position paper has four central goals. The first is to make a case for why it is important to understand psychological resilience to suicidal experiences generally, but also when working with people who have severe mental health problems, in particular, non-affective psychosis. The second goal is to look at what resilience means in a personal sense to those with psychosis who are also suicidal, and to further examine individual points of view about resilience in this population. The third goal is to examine approaches and models which can be used to guide ways of investigating resilience to suicidal thoughts and acts in people with psychosis. The fourth goal is to suggest the most effective ways of developing psychological approaches which cultivate, maintain and bolster psychological resilience to suicidal experiences, again in general, but more specifically so that people with psychosis have the opportunity to benefit.

Although there can be a tendency to view resilience as two static ends of a pole, namely that you do have resilience or you do not have resilience, it can be demonstrated that resilience is a resource which is dynamic and fluid depending on a multitude of personal and contextual factors, as opposed to being fixed and binary [1,2,3,4]. It has also been pointed out that resilience should not be viewed as residing just within the individual but that systems, for example, mental health systems, political systems, and educational systems, all contribute to resilience at individual, community and societal levels [5]. It should be noted that we use the term suicide and suicidality to include a range of suicidal experiences including thoughts, feelings, urges, compulsions, plans, intent and wishes to die.

### Why It Is Important to Examine Psychological Resilience to Suicidal Experiences

Suicidal experiences include suicidal thoughts, tendencies, urges, compulsions, plans, behaviours, attempts and death. Such experiences are a global concern with substantial suicide prevention measures being developed and implemented worldwide [6,7]. Understandably, considerable efforts have been invested from epidemiological, public health, and psychological endeavours into identifying which factors are associated with risk of dying by suicide. For example, being male, unemployed, having relationship problems, and having financial difficulties can all contribute to suicidal thoughts and acts, often cumulatively [8]. Psychologically, feeling isolated by others, not in control of mental and physical health problems, perceiving a future devoid of hope, and feeling trapped, can act as precursors to a range of suicidal experiences [9,10,11,12,13,14,15,16,17,18,19,20,21]. These types of antecedents are observable across samples of people with a range of severe mental health problems including those with non-affective psychosis [22]. For instance, childhood trauma was found to be a significant predictor of suicide attempts in individuals with a schizophrenia spectrum disorder [23]. In an investigation of the psychological precursors of suicidal experiences, hopelessness was a strong predictor of attempted suicide in people who were admitted to hospital psychiatric wards for the first time [24]. Furthermore, feelings of defeat and entrapment were identified as a mediator in the relationship between psychotic symptom severity and suicidal ideation in adults with non-affective psychosis [25].

It is important, not only to examine from numerous viewpoints the antecedents of suicidal experiences, but to also examine the obverse side of such antecedents, and investigate what enables people to have psychological resilience to suicidal thoughts and behaviours [1,26]. Suicide fatalities increase in those with severe mental health problems [27,28,29,30,31]. In approximately 90% of deaths recorded as suicide, those who died had severe mental health problems [32]. However, although there are elevated rates of suicidal thinking and acts in in people with severe mental health problems, including those with a diagnosis of schizophrenia, many do not experience suicidal thoughts, plans, or urges and, for those who do have these types of experiences, they, nevertheless, do not all attempt, or die by, suicide [33]. Research which concentrates on looking at resilience to suicidal thoughts and acts is relatively limited, especially compared to the plethora of work researching risk factors for suicidal thoughts and behaviours [34,35]. There needs to be a balance between investigating risk factors and investigating resilience factors with respect to understanding and addressing suicidal experiences [34]. There also needs to be a focus on using a variety of qualitative, quantitative, and mixed research methods in ways that promote convergence of findings to optimally understand resilience to suicidal thoughts and acts. Only by adopting this type of comprehensive, convergent and integrative approach does it seem possible to optimally develop evidence-based, efficacious, psychological interventions which build, maintain, and strengthen resilience to suicidal experiences.

## 2. The Meaning of Resilience and Personal Perspectives of Resilience for People with Psychosis Who Have/Had Suicidal Experiences?

Embracing the views and experiences of people with severe mental health problems with respect to understanding resilience to suicide is important for three inter-connected reasons. First, translational research with a remit of developing therapies and interventions which are genuinely meaningful to people, will benefit those individuals in a way that matters to them. Second, and relatedly, it is often necessary and important to understand the lived experiences of people with severe mental health problems if interventions which are effective and efficacious in the long and short term are to be advanced (Connell, Schweitzer, and King, 2015; Hamm and Leonhardt, 2016; Waite, Knight, and Lee, 2015). Third, a shared understanding of resilience is vital between mental health professionals and those who are suicidal and have severe mental health problems. This is because that shared understanding has the potential to enhance many levels of communication which has knock-on effects for countering distress associated with suicidal thoughts and acts and also in terms of developing effective, acceptable, and efficiently delivered multi-faceted psychological interventions (Collins, Peters, and Watt, 2011). This needs to be a priority.

### 2.1. The Meaning of Resilience as Expressed by People with Psychosis and Suicidal Thoughts/Acts

Psychological resilience has been studied across many domains. These domains have included, but are not limited to, neurobiological underpinnings [36]; children who have survived immense traumas including witnessing domestic violence [37,38,39]; war veterans and disaster survivors [40,41,42,43,44]; those in college and university education [45,46]; competitive sports performers [47]; refugees [48]; military personnel [49,50]; end-of-life care [51]; in front line health care staff [52,53]; in people with severe mental health problems [1,33,35,54,55]; and in community minorities with respect to ethnicity, sexual orientation, and gender identity [56,57,58,59]. One of the main issues with such diversity is that conceptualisations of resilience can become exceptionally broad and lacking in clarity [1,4,60]. For example, resilience has been equated with numerous psychological factors such as stoicism [61], being able to bounce back from adversity [62], coping [63], perseverance [64], having emotional stability [65], a defence mechanism [66], resistance [61] and more generally, in terms of personal resources which can act to counter the detrimental effects of internal and external negative stressors on severe mental health problems [67]. Sumskis and colleagues identified 14 definitions of resilience [68]. This kind of breadth means that the extent to which conceptualisations of resilience resonate with people with severe mental health problems may be questionable depending on their specific mental health problems and their numerous and diverse personal and living circumstances.

Based on a model of suicidal thoughts and acts, Johnson and colleagues attempted to identify key components of resilience to suicidality transdiagnostically [69]. These comprised perceptions of social support, feeling able to regulate and control emotions and feeling able to problem solve within a current situation [67]. However, we acknowledge that rather than imposing definitions and conceptualisations of psychological resilience on participants, one way forward is to openly ask people with mental health problems what resilience means to them. This type of approach tends to use qualitative methodologies. Qualitative work investigating resilience to suicidality is relatively sparse. That said, an interesting study asked 14 people with a diagnosis of schizophrenia who were currently well what resilience meant to them using qualitative interviews which were analysed with a phenomenological method. This study identified a number of factors relevant to resilience. Some were positive or supportive, such as “growing”, in which some control was felt over schizophrenia and the associated symptoms. Some factors were negative including feeling “lost”, in which some participants felt they were a victim or under pressure from others to change. Perhaps the most intriguing factors were those which could be both supportive and challenging simultaneously. For example, taking medication could be felt as challenging due to over-coming and managing the side effects but the upside of overcoming the challenge was sometimes seen as a therapeutic benefit. These sorts of dynamics occurred in the domains of family, work, social relationships, and within interactions between health professionals. The active decisions that participants made to embrace challenges were seen as fundamental to resilience. It was also documented in this study that participants sometimes felt that they were “going backwards” and that the process of “moving forward again” involved taking risks. Taking risks was viewed as an important part of resilience with the potential for support from friends, family, and mental health professionals whilst taking those risks [68].

Two points are worthy of note here. First, many people with schizophrenia can feel mentally unwell and yet still show resilience. Second, it is important to look at how resilience impacts suicidal thoughts and urges in people experiencing psychosis. Collaboratively exploring with people who are both suicidal and have psychosis what resilience means to them is not documented frequently in the literature. We attempted to examine this in people with non-affective psychosis who had experienced suicidal thoughts/acts in their lifetime in the context of resilience to negative stressors (*N* = 21). The most frequent negative stressors were mental health problems (schizophrenia, depression, suicidality) and illness/death of a friend or relative. It is worth noting that this study did not ask participants with psychosis about their experiences of resilience to suicide explicitly but rather about resilience to negative stressors. Of course, it has long been recognised that both internal stressors (e.g., hallucinations) and external stressors (e.g., relationship break-ups or bereavement) are precursors to suicidal thoughts/acts in people with severe mental health problems, such as, psychosis [27,30,32,69,70,71,72].

The first question that we asked participants was “What does resilience mean to you?” Thematic analysis was used to analyse the responses of participants [70]. Themes illustrated a dynamic spectrum of internal personal resilience resources related to participants’ own meaning of resilience. At one extreme end of the spectrum, resilience represented a passive acceptance of stressors: “You can come to terms with everything” [ID25] (p. 599). At the other end of the spectrum, resilience was expressed as an active response to stressors, often involving tackling, defying, or battling stressors “Energy to fight back” [ID19] (p. 600), “A sense of not being defeated” [ID18] (p. 600), and feeling able to cope “it would mean like how they cope with things” [ID1] (p. 600). Along this continuum there was also a sense of being able to resist or block negative stressors and being independent, for example, “……not caving in under pressure” [ID35] (p. 600); “I suppose it’s being able to stand up on your own two feet. I’m resilient that way, doing everything for yourself” [ID31] (p. 600). Participants expanded on ideas of active responses to adverse events which included working to stay positive “I have to condition myself every day, to think positive things rather than negative things” [ID32] (p. 600), and to be coherent “it was just like a matter of trying to stay rational” [ID27] (p. 600). Participants also highlighted the importance of external influences which helped their resilience, comprising social support, social reciprocity, and religion.

### 2.2. Understanding Resilience to Suicide from the Personal Perspectives of Those with Non-Affective Psychosis

Harris and colleagues [73] built on the previous study [70] with the aim of exploring personal perceptions of resilience to suicide explicitly, in people with non-affective psychosis. In-depth semi-structured qualitative interviews were conducted with 20 people who had psychosis, all of whom had experienced suicidal thoughts/behaviours in their lifetime with eight reporting suicide attempts. Four participants had suicidal thoughts at the time of their interview. The interviews were analysed using thematic analysis. Participants were asked about the pathways leading to resilience to suicidal thoughts/acts in their experience [73].

One of the key findings that came across very strongly was the sheer effort it took to be resilient on a day-to-day, hour-by-hour, basis when facing interactions between psychotic symptoms and suicidal thoughts. Resilience was, therefore, seen as an ongoing, and somewhat relentless, endeavour, which overlapped with attempting to stay well. For example, one participant stated, “…I’d have killed myself by now if I wasn’t resilient. […] I managed to keep going, even though I was hearing voices” [ID14, female] (p. 2–3). Another told us, “I’m trying to push myself to get better because the medicine will only do so much. You’ve got to help yourself as well” [ID11, male] (p. 3).

In order to develop and maintain resilience to suicidal thoughts and behaviours whilst experiencing psychosis, participants identified five key factors: i. understanding suicidal and psychotic experiences; “I wish I had more of an understanding on it [suicidal experiences] because, it’s like anything, if you understand that it’s wrong, in your mind you know it’s wrong” [ID17, female] (p. 3); ii. learning to live with/accept psychotic and suicidal experiences which could take time: “I sort of got used to it [the suicidal experience]. So, I must have thought to myself, you know, that this is just the way life is now. There’s nothing I can fucking do about it […] I just accepted it” [ID6, female] (p. 4); iii. feeling that there were reasons to live despite the relentlessness of suicidal and psychotic experiences: “I must have been [resilient], not to go through with it [the suicide attempt]. I must have felt there’s something worth living for” [ID4, male] (p. 4); iv being able to talk to people about suicidal thoughts and feelings: “I don’t have any tips and pointers on how to stop feeling suicidal, other than to talk to people… I know, it might upset people but it’s just, when you are having feelings of suicide, and you feel like ending it all, it’s just, it can be a very lonely place” [ID1, male] (p. 5); and v. interpersonal relationships with relatives and health care professionals “If the [suicidal] thought’s there but it’s not as strong, I can just get on with my day-to-day business because I’m used to it. If it’s stronger, then that’s when I’ll ring my CPN [Community Psychiatric Nurse]” [ID11, male] (p. 5).

### 2.3. The Meaning of Resilience: Conclusions

These qualitative findings are important scientifically from both theoretical and clinical frames of reference because: i. it was clear that the majority of the participants had their own understanding of what it meant to be resilient, which can be collaboratively built upon but also expands and refines a conceptual understanding of resilience to suicide more broadly; ii. participants highlighted the importance of not only embracing active aspects of resilience when feeling suicidal but also understanding more passive aspects which could be explored therapeutically; iii. we should understand, endorse, and validate the immense energy and effort that it can take for someone who is suicidal and also has psychotic experiences to simply get through their day; iv. many participants really seemed to strive to gain an understanding of the overlap between their psychosis and suicidal experiences which, again, seems hopeful from therapeutic perspectives; v. acceptance of the ways in which psychosis and suicidal experiences were interwoven came across as important, but was also countered by the idea that being able to resist or block the influences of negative stressors can be an effective way of being resilient for some individuals, perhaps highlighting a tension that we could be mindful of; and vi. working with people to try to identify their reasons for living seemed crucial, including acceptance and having personal values. This type of acceptance seemed to communicate a non-judgmental ability to actively live whilst experiencing suicidal thoughts/acts and behaviours, rather than simply being “ground down” into submission. One area which, perhaps, deserves further scrutiny from qualitatively designed studies is understanding ways in which medications feed into personal resilience. This, of course, may be best viewed on a continuum including ways in which managing medication seems to erode resilience but also to bolster it.

## 3. Mechanistic Approaches to Understanding Psychological Resilience to Suicide

We need to try to understand how resilience to suicidal experiences in those who have severe mental health problems, including psychosis, fluctuates within and between individuals using a multi-componential approach [34,35,67,74,75]. This understanding rests on identifying the psychological mechanisms which underpin resilience to suicidal experiences whilst also interacting with resilience to psychosis. Here, we focus on four models of resilience which may aid our understanding [76], namely, the unidimensional model, the two dimensional buffering model, the recovery model and the maintenance model. There is a fifth model of resilience which denotes immunity. However, along with others, we feel that it is implausible that anyone is immune to the negative effects of different types of stressors throughout their lives [4,77].

### 3.1. The Unidimensional Model: Two Poles of One Continuum

Resilience and risk factors are seen as at two ends of a pole in the unidimensional model. For example, a risk factor for suicidality may be feeling socially excluded whereas a resilience factor may be feeling socially connected and part of a community. There are two problems with this approach [35].

First, if a measure of social connectedness is found to be inversely related to a measure of suicidality, then it is sometimes concluded that social connectedness is a resilience factor. Indeed, social connectedness may form part of a resilience mechanism, but such a mechanism needs to be developed and tested. Unfortunately, there are many examples of this kind of assumption throughout the suicide literature which are too numerous to document.

The second and related problem is that not experiencing a suicide risk factor does not equate with a resilience factor. For example, hopelessness is a strong predictor of suicidal thoughts and acts including in those with psychosis [77]. However, not experiencing hopelessness is not synonymous with feeling hopeful about the future. Similarly, a negative perception of having serious financial problems is often one of the predictors of suicidal thoughts and acts [8]. To not have negative perceptions of having financial problems cannot be considered an aspect of resilience without further examination because: i. the financial problems may have resolved perhaps because someone close offered a loan, and/or ii. the relationship between negative perceptions of financial pressures and suicidality may have been buffered and weakened by other personal resources, such as finding other ways to alleviate financial pressures. In this case, the latter situation may represent a source of resilience which could be further explored.

### 3.2. The Two-Dimensional, Buffering, Model: Two Independent Dimensions

The two-dimensional approach to understanding resilience expands on the unidimensional approach. In the two-dimensional approach, risk factors and resilience factors are seen as independent dimensions. So, if a risk factor is present, such as intense stressors, it does not imply a lack of resilience. Indeed, if we go back to some of our qualitative findings, participants with psychosis who also had suicidal thoughts and acts were still able to get through their day, even though it took the most immense amount of effort for individuals to exist with mental health problems, and, in some cases they did this whilst accepting what they were going through and feeling that there were still reasons to live [73].

The buffering model takes the idea of two dimensions a little further. Resilience as a buffer examines the extent to which resilience factors can counter negative internal stressors (e.g., hallucinations) and external stressors (e.g., late disability payments) by weakening associations between such stressors and a negative outcome, such as, suicidal thoughts and behaviours [34,35]. To take a concrete example, a strong and positive association may be found between experiencing financial pressures and, because of those pressures, developing suicidal thoughts, but feeling meaningfully connected to others may counter, weaken or buffer, that association between financial pressures and suicidal thoughts and/or plans [78].

One cross-sectional study which recruited people who were suicidal and had psychotic symptoms found that high levels of resilience weakened the association between perceptions of stressful life events and suicidal thoughts and acts [67]. Another prospective study over a 3-month time span reported that over that time, defeat and entrapment predicted suicidal thoughts/acts only when psychotic experiences and the ensuing distress were high, and resilience was low. When resilience was low, this relationship between defeat, entrapment and suicidal thoughts/behaviours was amplified [79]. Hence, individuals reporting low levels of resilience may be more likely to experience suicidal thoughts and behaviours because of defeat, entrapment, psychosis, and the associated distress. Resilience in both studies was conceptualised as positive self-appraisals of emotional coping, social support and situation specific problem solving based on a theoretical model of psychological pathways to suicidal thoughts and acts [67]. It is important to note that many studies in the area of understanding pathways to suicidal experiences and pathways to resilience to suicide are cross-sectional. When examining the role of buffers to pathways to suicidality, prospective qualitative and quantitative designs, including those that are micro-longitudinal [80] are to be encouraged although not exclusively so [81].

### 3.3. The Recovery Model

There is a large and growing literature examining the concept and mechanisms of recovery in people with severe mental health problems including psychotic symptoms [82,83]. Nevertheless, it has been pointed out that the idea of recovery often lacks a definitional consensus which can sometimes challenge scientific and clinical progress [84,85,86,87,88].

Many conceptualizations of recovery have followed a medical model in which: i. negative events and negative internal and external stressors were identified; ii. the subsequent detriment to functioning caused by these events and stressors was assessed; and then iii. the degree to which these “deviations from normality” could be rectified signifying a return to normality or remission from the perceived ‘medical problem’ was determined [88,89]. The medical problem was defined in terms of symptoms. Health was defined as an easing of those symptoms [89,90,91,92].

The idea of recovery, derived from more contemporary frameworks, focuses on how recovery can be best defined and understood from the personal perspectives of experts-by-experience (sometimes referred to as consumers) and not exclusively from psychiatric symptoms nomenclature. That is, a distinction is made between clinical recovery and personal recovery [83]. This grounding of recovery on the personal perspectives of people who are experts-by-experience is considered fundamental if advances in understanding recovery processes and mechanisms are to be made in ways that make a meaningful difference to experts-by-experience, scientists, clinicians and policy makers [93,94,95,96,97].

Personal recovery embraces and explores experiences of personal growth as a result of coming through, or better still, being able to simply ‘be with’ severe mental health problems [68,95,96,97,98]. There are extensive evidence-based examples of personal recovery in people with psychosis [82,83,88]. Yet, the question arises concerning the extent of the evidence for personal recovery in people experiencing both psychosis and suicidal thoughts/acts in tandem. From one of our qualitative studies looking at the experiences of resilience expressed by people who had suicidal thoughts/acts the following quote illustrates the way that resilience was seen to develop from coming through extremely difficult personal problems [73]: “[Resilience] is developed. Unless you experience some deep problems, you don’t actually get to find out whether or not you’re resilient” [participant 14, woman] (p. 4). In the same study, another of our participants brought home to us the idea that resilience can encapsulate simply ‘being with’ and accepting suicidal thoughts with other aspects of life carrying on in parallel with those suicidal thoughts: ‘…what you do is accept the fact that it [the suicidal thought] exists […] you just have it as an existing part of your mind’ [participant 8, man] (p. 4). A different participant underscored that for them acceptance was key: ‘I sort of got used to it [the suicidal experience]. So, I must have thought to myself, you know, and accepting this is just the way life is now. There’s nothing I can fucking do about it […] I just accepted it’ [participant 6, woman] (p. 4).

Research that aims to understand recovery based on the views of a diverse range of people experiencing psychosis is expanding [93,94,95,96,97,98,99,100,101,102,103]. As we have seen from some of the qualitative work that we have presented, it is not necessarily the case that the only way of finding meaning in life whilst having mental health problems is to escape from those problems [73]. It is potentially enlightening, especially from a therapeutic point of view, that some people have found that for them, suicidal experiences can co-exist in parallel with other aspects of their lives.

### 3.4. The Maintenance Model

The maintenance model is most closely allied with the positive psychology movement, one of the goals of which is to understand how people prosper and grow over time and across different realms of their lives [102,103,104,105]. Being founded on the Psychological Flexibility Model [105], the Broaden and Build theory [104,105], and the Self-Determination Theory [106,107], this model describes a set of abilities which enable people to maintain a sense of positivity when facing negative experiences, stressors, and health conditions some of which are chronic and occur over long time periods. Being able to identify and engage with meaningful activities and values, despite significant adversity, is central to the maintenance model which was developed in the context of coping with chronic pain [88]. An important point is that this approach to resilience is not characterized by recovery nor a resolution/partial resolution of health problems due to buffering factors. Embracing personally meaningful activities and developing values which are important to an individual may be observed even when there is no improvement in health problems. One suggestion is that aspects of these kind of positive thoughts and behaviours may serve to deflect foci away from health problems in addition to strengthening psychological flexibility and the experience of positive emotions together with monitoring perceptions about the extent to which important personal needs have been both identified and fulfilled [88].

It can be argued that qualitative methodologies are best suited to exploring ways that people with psychosis who are suicidal are able to identify and enact personal values in the face of severe mental health problems, whilst also accepting those mental health problems at least to some extent. In addition, some individuals have explained to us the means by which they have been able to find ways of distracting themselves from their mental health problems as these quotes illustrate [33,73]: ‘I like cooking. It just takes your mind away […] you need to distract yourself in some way’ [participant 2, man] (p 5); ‘It [music] takes the edge off it [the suicidal thought] but doesn’t completely get rid of it… it’s easier to manage’ [participant 3, man] (p. 5).

### 3.5. Mechanistic Approaches to Understanding Psychological Resilience to Suicide: Conclusions

A narrative systematic review [26] including 31 publications investigated resilience in adults with mental health problems. The review highlighted two over-arching ways of conceptualising psychological resilience. The first was as a process and the second was as personal characteristics or resources of an individual. There were three concepts: i. immunity, such as, being able to resist the effects of stressors; ii. bouncing back, defined as returning to a state of cognitive-emotional-behavioural stability; and iii. growth in which positive transformation was harnessed as a result of adversity. Characteristics of individuals included: i. having personal resources and ii. having social resources, both of which were seen as having the potential to build and sustain resilience.

Perhaps, thinking about resilience as a set of interacting processes which includes the development of multi-dimensional personal resources but also encompasses accessibility to, together with a feeling of personal ownership of social, practical and societal resources, offers the greatest potential for nurturing resilience across different levels, e.g., societal, community and individual. In other words, resilience may be affected by a range of modifiable personal, psychosocial and political factors [74,108,109]. Of the four models identified; buffering (two-dimensional), recovery and maintenance, seem to offer the most potential with respect to investigating resilience to suicidality. That the psychosocial factors focused on in these models are adaptable, perhaps, offers pathways for developing psychological approaches and techniques which can maximise and sustain resilience to suicidal experiences.

## 4. Psychological Interventions Which May Support and Help Develop Resilience to Suicidal Experiences

There is an expanding literature which recognises the value of developing psychological and related therapies that not only work on reducing suicidal thoughts/behaviours but can also act to increase and/or consolidate individuals’ resilience to suicidality in particular [2,34,109,110,111,112]. For example, in one small study with Nepali women who had histories of suicidal experiences, ten sessions of an adapted form of Dialectical Behaviour Therapy were offered [113]. There were improvements in emotion regulation skills, scores on a resilience measure, and marked reductions in suicidal ideation over time. With a focus on depressive symptoms, another study used a fully automated CBT program delivered by the internet [114]. Decreases in self-reported depression scores over time, including the one item that assessed suicidal ideation, and increases in scores on a measure of resilience were reported. There are few examples of therapies designed to examine resilience together with suicidal experiences in people with psychosis. That said, some studies have the potential to do so. For instance, one pilot trial, the results of which were not yet known at the time of writing, aims to examine the transition from acute to ongoing care for people with psychoses (bipolar disorder, schizophrenia, schizoaffective disorder) who are actively suicidal, using a bespoke suicide-focused SafeTy and Recovery Therapy (START) comprising four weekly therapy sessions [115]. The results, which centre on acceptability, feasibility and effectiveness, seem to represent an intriguing and promising start to this work.

We have suggested that psychological therapies which aim to most effectively ameliorate suicidal experiences in people with psychosis should be focused on those experiences and, most importantly, on the underlying mechanisms which give rise to those suicidal experiences, rather than paying attention just to psychiatric symptom reduction [92,116,117,118,119,120,121,122]. This approach is somewhat different from seeing whether a therapy is associated with less suicidal experiences together with increases in scores on a measure of resilience over time because this type of work does not probe the underlying mechanisms. Based on the Schematic Appraisals Model of Suicide (SAMS) [67], we have now piloted our theoretically derived suicide focused therapeutic cognitive behavioural intervention (CBSP) in people with psychosis living in the community [117], in prisoners [119,120], and in those residing in in-patient psychiatric wards [121]. Results have been encouraging: suicidal thoughts and behaviours have been found to be reduced in those in the treatment arm (Therapy plus Treatment as Usual) compared to the control arm (Treatment As Usual alone) of these studies. However, this work, thus far, has been at the level of pilot randomised controlled trials (RCTs).

The SAMS was formed not only to help elucidate the mechanisms leading to suicidal thoughts and acts but also to scaffold the development of testable mechanisms, which are central in the pathways to resilience to suicidal experiences. Again, we suggest that it is only by understanding these resilience mechanisms that therapeutic approaches which build resilience to suicidality can be optimally formulated and offered to people with severe mental health problems. Grounded in empirical work with people, including but not limited to schizophrenia [67,73,79,116,117,118,120,121], one of our tenets is that positive appraisals of emotional regulation, social support and situation-focused problem solving are key in resilience mechanisms to suicidality. We aim to determine the extent to which these positive appraisal components of resilience will be improved by suicide-focused cognitive therapy over time and whether strengthening those components will counter suicidal thoughts and acts using statistical modelling techniques. Consequently, we are testing these predictions in a large multi-site RCT, called CARMS (Cognitive AppRoaches to coMbatting Suicidality) with three time-points of baseline, and 6 and 12-month follow-up periods [118]. Crucially, we have recruited individuals with non-affective psychosis who were suicidal in the 3 months prior to being included in the trial. This approach is most closely allied to the buffering (two-dimensional) model of resilience. However, it also includes nested qualitative work which will allow us to gain deeper insights into both the pathways to suicidal experiences and the pathways to resilience to suicide.

### Future Directions with Respect to Psychological Interventions

The work from qualitative studies is exciting with respect to developing models of resilience to suicidal experience in people with psychosis. One of the key findings from qualitative work is the sheer ongoing effort that it can take for people with hallucinations and delusions to get through their day [73]. Therapeutically, expressing a genuine understanding or genuine desire to understand more about what this effort entails seems key. A second key finding is the immense struggle that many people with severe mental health problems who are also suicidal face with respect to different types of stigma, for instance self-stigma, stigma from mental health professionals, stigma from relatives and/or family members, stigma in the work place and, more broadly, stigma at a societal level [33,73,89,123]. Embedding an appreciation of this effort into therapeutic approaches on a regular basis might convey positive reinforcement for all the techniques and strategies that individuals are using to live their lives, often spontaneously, in addition to communicating immense positive regard to individuals on the part of the therapist. A feeling that the therapist is committed to “being with” a client seems important, as many people with severe mental health problems can feel abandoned by both mental health services and relatives or friends. Relatedly, an invitation by the therapist to the client to explore any values that they hold may feel welcome.

Although somewhat limited currently, there is some evidence that psychological therapies which do not centre around talking (but nevertheless do not exclude it) may be beneficial to those who are suicidal and have severe mental health problems. For example, manualised phenomenological art therapy was offered to people with moderate to severe depression. The design was a two-armed, small-scale RCT. At the 6-month follow-up time point, a measure of suicidal ideation indicated improvement, as did a measure of self-esteem [124]. An interesting qualitative study explored the role of dance in “body memory” in people who were bereaved as children/adolescents. Two themes were highlighted which made connections between bodily sensations and emotions. The first was the relationship between the body and emotions as the person moved towards resilience. The second pinpointed relationships between bodily tensions and experienced emotions [125]. In a recent case study, 12 sessions of dance movement across a three-month period were attended by a professional artist who had schizophrenia, with drawing and painting being used as part of those therapeutic sessions. The sessions had a fluidity, with one session feeding into another through a connection between bodily experiences, art, and the expressed reality of the artist. Compared to baseline, measures of symptoms of psychosis were low, perceptions of coping had improved, potentially indicating resilience, and bodily movement had also improved in quality [126]. Being able to offer individuals who are suicidal and have severe mental health problems an eclectic approach of, preferably, resilience-to-suicide-focused talking therapies (e.g., CBSP, ACT, DBT) together with different types of bodywork presents an exciting future avenue.

## 5. Conclusions

In conclusion, examining resilience to suicidal experiences in people with severe mental health problems shows immense promise, scientifically, clinically, and in terms of developing health policies which are feasible to deploy. Such initiatives require a focus on involving experts-by-experience in resilience to suicide endeavours, not only in research projects but in political conversations spanning communities and societies.

## Data Availability

Not applicable.

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
