# Peer review of "Psychological Resilience to Suicidal Experiences in People with Non-Affective Psychosis: A Position Paper"

_ijerph, 2022, doi:10.3390/ijerph19073813_

Round 1

Reviewer 1 Report

This is an interesting position paper.  Research into affective disorders and suicide tend to focus on the risk factors, but less on the protective factors.  Community assets or resources are often overlooked when studying resilience in groups experiencing extreme stress. The paper sets out well it goals.  Few papers have attempted to use qualitative data to define resilience to suicidality, especially in those with psychosis. 

Some comments and questions:

  1. On line 173-178, the researcher mention that in the 21 qualitative interviews, the participants were not asked directly about resilience to suicide, but about resilience to negative stressors. These sentence might be best if moved up to about line 155-156 before the analysis of responses from participants so readers know what exactly the responses in related to.
  2. Reading the statements made by participants in the two samples of 21 and 20, it is hard not to wonder whether the conditions were being managed with medications or whether the medications were not being taken, or were not working. Was there any differences noted by participants about how medications helped them with resiliency or were medications and their side-effects another factor that required more resilience. It probably depends on the medication and what it was being used for. 
  3. Was acceptance as a form of resilience related more to leaned helplessness or true resilience? It seems that acceptance was more about acknowledging suicidal thoughts without judging them or giving them control over one’s life. Is there any additional information you can add about the experience of “acceptance”?
  4. Given the literature on suicide, it seems as if those experiencing suicidality but without a psychosis also experience these same approaches to resilience and coping. It must be true that symptoms such as hallucinations and a possibly reduced ability to function at times adds far more stress requiring resilience, yet not all in these situations have suicidal ideation.  The comment about finding a reason to live is an important factor and has been addressed by many in psychiatry. This would make their experiences even more informative to addressing suicide and mental health challenges, as is stated in the first few sentences of section 2.3. How should therapies prioritize resilience in treating these individuals?
  5. The explanation of the four models of resilience is useful for providing a framework for the preceding sections of the paper.
  6. The closing paragraph discussing diverse approaches to therapy in those with serious mental health challenges is a satisfying way in which to summarize the state of the art in therapy.

Author Response

Please also see uploaded file for responses to reviewer 1.

Comments and Suggestions for Authors

We would like to thank the reviewers for their thoughtful and helpful comments. Our responses are presented below to each point raised.

Reviewer 1

This is an interesting position paper.  Research into affective disorders and suicide tend to focus on the risk factors, but less on the protective factors.  Community assets or resources are often overlooked when studying resilience in groups experiencing extreme stress.

  • We are extremely appreciative that the reviewer has understood the gap in the area of suicide research in that resilience to suicidal experiences has not yet been fully examined or explored. We resonate with the reviewer’s point about community resources.

The paper sets out well it goals. 

  • We are pleased that the goals seemed clear.

Few papers have attempted to use qualitative data to define resilience to suicidality, especially in those with psychosis. 

  • We are delighted for acknowledgment of the role that qualitatively designed research can play in understanding resilience to suicidal experiences.

Some comments and questions:

  1. On line 173-178, the researcher mention that in the 21 qualitative interviews, the participants were not asked directly about resilience to suicide, but about resilience to negative stressors. These sentence might be best if moved up to about line 155-156 before the analysis of responses from participants so readers know what exactly the responses in related to.
  • We agree with this point and have moved the sentence up, as advised, to line 156.

  1. Reading the statements made by participants in the two samples of 21 and 20, it is hard not to wonder whether the conditions were being managed with medications or whether the medications were not being taken, or were not working. Was there any differences noted by participants about how medications helped them with resiliency or were medications and their side-effects another factor that required more resilience. It probably depends on the medication and what it was being used for. 
  • We felt that this was a very apposite point. We did not explicitly ask participants about their perceptions about the role of resiliency in managing medications including their side effects. Hence, we have added these statements to highlight this point “One area which, perhaps, deserves further scrutiny from qualitatively designed studies is understanding ways in which medications feed into personal resilience. This, of course, may be best viewed on a continuum including ways in which managing medication seems to erode resilience but also to bolster it.” Line 250.

  1. Was acceptance as a form of resilience related more to leaned helplessness or true resilience? It seems that acceptance was more about acknowledging suicidal thoughts without judging them or giving them control over one’s life. Is there any additional information you can add about the experience of “acceptance”?
  • This is a very good point. We do not have any data, i.e., quotes, which speak directly to this issue. Therefore, we have expanded a little about what it might mean to accept “This type of acceptance seemed to communicate a non-judgmental ability to actively live whilst experiencing suicidal thoughts/acts and behaviours, rather than simply being ‘ground down’ into submission.” Line 248.

  1. Given the literature on suicide, it seems as if those experiencing suicidality but without a psychosis also experience these same approaches to resilience and coping. It must be true that symptoms such as hallucinations and a possibly reduced ability to function at times adds far more stress requiring resilience, yet not all in these situations have suicidal ideation.  The comment about finding a reason to live is an important factor and has been addressed by many in psychiatry. This would make their experiences even more informative to addressing suicide and mental health challenges, as is stated in the first few sentences of section 2.3. How should therapies prioritize resilience in treating these individuals?
  • We are grateful to the reviewer for encouraging us to elaborate a little here. Hence we have added “Therapeutically, expressing a genuine understanding or genuine desire to understand more about what this effort entails seems key.” Line 475.

The importance of a truly reflective desire for a therapist to embrace stigmatising experiences in the context of expressing positive regard we have included in Line 481.

We have added the point that rather than simply attempting to tease out ‘reasons for living’ from the perspectives of a client, that it may be more helpful to i. simply ‘be there’ for the client because they may feel that they have been abandoned, especially when feeling suicidal, and ii. to explore the values that a client has ‘in-the-moment’ as follows: “A feeling that the therapist is committed to ‘being with’ a client seems important as many people with severe mental health problems can feel abandoned by both mental health services and relatives or friends. Relatedly, an invitation by the therapist to the client to explore any values that they hold may feel welcome.” Line 484.

  1. The explanation of the four models of resilience is useful for providing a framework for the preceding sections of the paper.
  • We were pleased that this was useful.

  1. The closing paragraph discussing diverse approaches to therapy in those with serious mental health challenges is a satisfying way in which to summarize the state of the art in therapy.
  • We were delighted that the diverse approaches to therapy in the context of resilience to suicidal experiences seemed helpful to the reviewer and, hopefully, helpful to readers.

Reviewer 2 Report

Thank you for allowing me to review this manuscript regarding psychological resilience to suicidal experiences in people with non-affective psychosis. Given the elevated risk of suicidal behaviour in patients with psychosis, and the need to provide adequate psychosocial support, this review could be a timely publication. However, sadly, the manuscript entails crucial shortcomings.

The review has five goals / research questions. The rationale for these questions has not been presented. The review has no introduction that provides a background or a motivation for these goals or research questions.

There is also no ‘Methods’ section. It is unclear how the literature has been searches, selected and analyzed. There is also no quality appraisal of the literature. Hence, it appears that the manuscript presents the authors’ personal views and opinions rather than an analysis and synthesis of the literature.

I would recommend conducting a systematic review of the literature according to PRISMA guidelines. In its current form, I see little merit in publishing the manuscript, for the readers and authors alike.

Author Response

Please also see uploaded doc for responses to reviewer 2.

Comments and Suggestions for Authors

We would like to thank the reviewers for their thoughtful and helpful comments. Our responses are presented below to each point raised.

Reviewer 2

Thank you for allowing me to review this manuscript regarding psychological resilience to suicidal experiences in people with non-affective psychosis. Given the elevated risk of suicidal behaviour in patients with psychosis, and the need to provide adequate psychosocial support, this review could be a timely publication. However, sadly, the manuscript entails crucial shortcomings.

The review has five goals / research questions. The rationale for these questions has not been presented. The review has no introduction that provides a background or a motivation for these goals or research questions.

There is also no ‘Methods’ section. It is unclear how the literature has been searches, selected and analyzed. There is also no quality appraisal of the literature. Hence, it appears that the manuscript presents the authors’ personal views and opinions rather than an analysis and synthesis of the literature.

I would recommend conducting a systematic review of the literature according to PRISMA guidelines. In its current form, I see little merit in publishing the manuscript, for the readers and authors alike.

  1. We were disappointed to read the comments of Reviewer 2. We believe that Reviewer 2 has misunderstood the type of paper that we have written. Hence, we would like to make four points in an attempt to clarify this misunderstanding:
  2. First, this is a Position paper as we state at the beginning of the manuscript. It is not a systematic review which would be structured and approached traditionally, as the reviewer suggests, with a Background, a Method section and so forth. We have now tried to make clear that it is a Position paper in the Title.
  3. Second, and relatedly, the reasons why we have chosen to present work in the form of a Position paper that is relevant to the development and maintenance of resilience to suicidal experience in those with severe mental health problems is that published research in this area is sparse, albeit growing. The extant ‘critical mass’ of relevant literature does not yet allow us to synthesise it using systematic review methodologies which facilitate the integration of findings convergently using quantitative, qualitative and mixed methods designs.
  4. Third, and again relatedly, therapeutic advances which directly address suicidal experiences in people with severe mental health problems including psychosis are minimal. Our work does exemplify this approach, that is, of addressing suicidal experiences explicitly as opposed to addressing mainly psychiatrically defined symptoms. In developing our therapeutic approach, which is based on scientifically testable models of the pathways to suicidal thoughts/acts, we perceived it as important to ensure that those models allowed pathways which seem to lead to resilience to suicide to be explored further. Examining pathways which underpin resilience to suicide in such models still requires massive expansion. Hence, the appropriateness of a Position paper which can hopefully be part of encouraging this kind of expansion.
  5. Finally, we have tried to emphasise that psychological models of resilience to suicidal experiences should be empirically testable. In our position paper, we have also tried to emphasise that a creative use of, and examination of, evidence incorporating many different therapeutic approaches seems scientifically and clinically exciting. We hope that this kind of positive approach can have a ‘real impact’.
